# iPSC-Cardiomyocyte Models of Brugada Syndrome—Achievements, Challenges and Future Perspectives

**DOI:** 10.3390/ijms22062825

**Published:** 2021-03-10

**Authors:** Aleksandra Nijak, Johan Saenen, Alain J. Labro, Dorien Schepers, Bart L. Loeys, Maaike Alaerts

**Affiliations:** 1Center of Medical Genetics, Faculty of Medicine and Health Sciences, University of Antwerp and Antwerp University Hospital, 2650 Antwerp, Belgium; aleksandra.nijak@uantwerpen.be (A.N.); dorien.schepers@uantwerpen.be (D.S.); bart.loeys@uantwerpen.be (B.L.L.); 2Department of Cardiology, Faculty of Medicine and Health Sciences, University of Antwerp and Antwerp University Hospital, 2610 Antwerp, Belgium; johan.saenen@uza.be; 3Department of Biomedical Sciences, Faculty of Pharmaceutical, Biomedical and Veterinary Sciences, University of Antwerp, 2610 Antwerp, Belgium; alain.labro@uantwerpen.be; 4Department of Basic and Applied Medical Sciences, Faculty of Medicine and Health Sciences, Ghent University, 9000 Ghent, Belgium; 5Department of Human Genetics, Radboud University Medical Center, 6525 Nijmegen, The Netherlands

**Keywords:** brugada syndrome, inherited cardiac arrhythmia, induced pluripotent stem cells, iPSC-cardiomyocytes, electrophysiology

## Abstract

Brugada syndrome (BrS) is an inherited cardiac arrhythmia that predisposes to ventricular fibrillation and sudden cardiac death. It originates from oligogenic alterations that affect cardiac ion channels or their accessory proteins. The main hurdle for the study of the functional effects of those variants is the need for a specific model that mimics the complex environment of human cardiomyocytes. Traditionally, animal models or transient heterologous expression systems are applied for electrophysiological investigations, each of these models having their limitations. The ability to create induced pluripotent stem cell-derived cardiomyocytes (iPSC-CMs), providing a source of human patient-specific cells, offers new opportunities in the field of cardiac disease modelling. Contemporary iPSC-CMs constitute the best possible in vitro model to study complex cardiac arrhythmia syndromes such as BrS. To date, thirteen reports on iPSC-CM models for BrS have been published and with this review we provide an overview of the current findings, with a focus on the electrophysiological parameters. We also discuss the methods that are used for cell derivation and data acquisition. In the end, we critically evaluate the knowledge gained by the use of these iPSC-CM models and discuss challenges and future perspectives for iPSC-CMs in the study of BrS and other arrhythmias.

## 1. Introduction

Brugada syndrome (BrS) is an inherited cardiac arrhythmia characterized by a typical pattern of ST-segment elevation on the electrocardiogram (ECG) and an increased risk of ventricular fibrillation and sudden cardiac death (SCD). It accounts for 20% of SCD in individuals without structural heart disease. [1]. In 20–25% of BrS patients, loss-of-function mutations are identified in the *SCN5A* gene, which encodes the α subunit of the cardiac sodium channel Na_v_1.5 [2]. To date, more than 20 other genes have been associated with this oligogenic disease with variants impairing specific ion channels or their accessory proteins involved in the cardiac action potential (AP) (reviewed in [2,3]). Still, three quarters of the diagnosed BrS patients remain without an implicated genetic variant [4]. Importantly, no comprehensive clinical and cellular studies have confirmed most of the candidate gene associations. A recent burden analysis and re-evaluation of reported genes only classified the *SCN5A* gene as demonstrating definitive evidence as a cause for BrS [5,6].

Concerning pathophysiology, currently three major hypotheses aim to explain the electric abnormality in BrS, namely the repolarization, depolarization and neural crest models [7,8,9,10,11,12,13]. In short, the characteristic ECG changes are respectively explained by (1) transmural dispersion of repolarization in the right ventricle between the endocardium and epicardium; (2) delayed depolarization due to conduction slowing and presence of subtle structural abnormalities in the right ventricular outflow tract (RVOT) and (3) abnormal cardiac neural crest cell migration, cell–cell communication and the development of the RVOT. Despite their differences, these models agree that the major region of pathology is the anterior RV and RVOT and that minute tissue architectural changes or cellular uncoupling effects play a role. Moreover, they are not mutually exclusive and could act in combination. Further studies are required to deepen our understanding of these mechanisms, how they lead to sustained ventricular arrhythmias and link them with the molecular changes underlying BrS.

BrS is well-known to display reduced penetrance and variable expressivity, characterized by a wide range of severity from a life-time asymptomatic status to syncopes and SCD at a young age, even in patients with the identical familial mutation [1]. This clearly hampers patient risk stratification and management. Both environmental and genetic disease modifiers play a role in this variability [14] and over the past years, BrS has been recognized as a more complex disease, with the involvement of multiple common and rare genetic variants acting in concert in its etiology [15,16,17,18,19]. Current application of gene panels, whole exome sequencing (WES) and whole genome sequencing (WGS) in (familial) patients will continue to identify variants potentially involved in BrS, though most of these remain “variants of uncertain significance” (VUS) and without functional studies, a gap remains to translate potential genotype–phenotype correlation into clinical practice.

To further elucidate the oligogenic architecture and pathogenic mechanisms of BrS, understand the functional effects of candidate pathogenic variants at the cell and tissue level and interpret the causality of VUS, representative functional disease models are highly needed. If successful, these models can be applied to test therapeutic approaches. Since currently no proper pharmacological treatment for BrS is available, the implantation of a cardioverter defibrillator (ICD) is the only effective preventative treatment for symptomatic patients. Recently, radio frequency ablation of the arrhythmogenic substrate, mostly in the epicardial surface of the RV(OT), is emerging as a potential alternative [20,21,22,23].

The challenge for functional modelling of cardiac arrhythmias, including BrS, is obtaining tissue specific material. Cardiac biopsies are considered too invasive and the obtained cardiomyocytes (CMs) have a short lifespan in vitro, making them not readily available at sufficient quantities, though some studies on native human CMs have been performed [24,25,26,27,28]. This issue has been addressed by the study of murine, canine, zebrafish or pig cardiomyocytes and cardiac tissue, which offered interesting insights in BrS pathophysiology (extensively reviewed in [29]). The major drawback of those systems are the species-specific differences in physiology, which impacts the translation of the results into the human clinical setting. As another alternative, non-cardiac cellular models have been employed (mostly human embryonic kidney (HEK)293, Chinese hamster ovary (CHO) cells or *Xenopus laevis* oocytes) to transiently overexpress human cardiac proteins, which enables one to study the function of individual channel complexes and specific mutations (extensively reviewed in [29]). The main shortcoming of these transient expression systems is their disability to mirror the complex CM physiology, as they lack the structural morphology and multi-ion channel environment of native CMs. In addition, results also tend to differ according to the cell-type used. The discovery of induced pluripotent stem cell (iPSC) reprogramming by Takahashi and Yamanaka in 2006 [30], and their subsequent differentiation to functional induced pluripotent stem cell (iPSC)-derived cardiomyocytes (iPSC-CMs) provided an interesting new alternative to the field of cardiac disease modelling. iPSC-CMs provide the (near) complete repertoire of ion channels and accessory proteins expressed in native human CMs, and currently represent the closest resemblance to these cells [31,32]. Since iPSC-CMs can be obtained directly from the patient, they carry the patient’s exact genetic background, including potential modifiers influencing the phenotype.

To date, multiple studies on cardiac arrhythmias using iPSC-CMs have been reported, proving the beneficial impact of this model in the field. A Pubmed literature review, performed in December 2020, identified thirteen articles reporting findings from iPSC-CM models for BrS. In this review, we will provide a comprehensive synthesis of these reports, with a focus on the electrophysiological findings and advances in the field of BrS pathogenesis. We will first discuss the methods used for the derivation of the cell models as well as the techniques used for phenotypic investigations.

## 2. Methods for Derivation of iPSC-CM Models

### 2.1. Fibroblast Reprogramming Protocols

The first step in the generation of patient-specific iPSC-CMs is the generation of iPSC lines from a donor sample such as a skin biopsy, or the more easily accessible blood or urine samples [30,33,34]. In all of the reviewed articles, the traditional approach with iPSC reprogramming from skin fibroblasts was applied (Appendix A). Several delivery methods to introduce the Yamanaka transcription factors can be used. Lenti- or retro-viral vectors are very efficient but have the major drawback of viral genome integration into the host genome [35]. To overcome that issue, non-integrating episomal vectors or Sendai virus vectors are increasingly used [36,37]. In the generation of the published iPSC-CM BrS models, we observed an equal distribution of the delivery methods, with 9/17 (53%) of the cell lines generated using integrating vectors and 8/17 (47%) using a non-integrating approach (Appendix A), but with the non-integrating methods indeed overrepresented in the more recent articles.

### 2.2. Cardiomyocyte Differentiation

Over the past 10 years, a constant progress in cardiomyocyte differentiation protocols has been made. From initial protocols employing 3D aggregate embryoid body (EB) differentiation with BMP4 and activin A, disease modelling moved to monolayer-based approaches with Wnt pathway modulation using the timed addition of small molecules [38,39,40,41,42]. In the latter, to start the differentiation process the Wnt pathway is indirectly activated through the inhibition of glycogen synthase kinase 3β (e.g., the addition of CHIR99021). Subsequently, after culturing for two or three days, a Wnt pathway inhibitor is added to the culture medium (e.g., Wnt-C59 or IWR1). By day seven or eight, the cultured cells start beating spontaneously, which is one of the first signs of differentiation towards CMs [43].

Though both EB and monolayer-based methods provide decent efficiency in the derivation of functional iPSC-CMs, the monolayer approach provides the most optimal conditions for the diffusion of the differentiating factors and highest differentiation efficiency. This is also reflected in the literature, as the majority of iPSC-CMs were derived through monolayer-based protocols (9/15) (Appendix A). These protocols were shown to produce mainly ventricular cardiomyocytes, with few atrial and nodal cells present, which is favorable in view of the ventricular origin of BrS pathology.

Nonetheless, the major and well-known disadvantages of iPSC-CM models are the immaturity of the cells, as well as heterogeneity of the cell culture obtained during the differentiation procedure [21,32,44,45]. Maturation can be improved through long-term culture, mechanical stretching or electrical stimulation, via the application of maturing agents such as triiodothyronine hormone (T3) and glucocorticoid hormone, 3D-culture, specific miRNAs or co-culture with human mesenchymal cells [46,47,48,49,50,51,52,53,54]. In most of the published iPSC-CM BrS models, no specific maturation strategies were used, just a slightly longer culture of at least 30 days before functional testing was performed (except for [55] with at least 19 days). Only de la Roche and colleagues used the cultivation of the iPSC-CMs on a stiff matrix to improve maturation [56]. Further purification of the cell culture with the selection of properly differentiated cardiomyocytes can be obtained with metabolic enrichment approaches, including simple glucose starvation, or substitution of glucose in culture media with lactic acid or fatty acids, to force the switch to a non-oxidative metabolism in the cell culture [57,58,59]. In 6/13 reports, culture heterogeneity was not addressed, and in the other 7/13 glucose starvation or puromycin or lactate treatment was implemented (Appendix A).

## 3. Phenotyping Methods of iPSC-CMs

Since BrS is associated with ventricular pathology [60], analysis of the electrical activity of ventricular cardiomyocytes is the main focus in BrS iPSC-CM modelling and it is important to understand its basis.

As indicated in Figure 1, the ventricular action potential (AP) reflects a sequential activation and inactivation of ion channels, conducting inward depolarizing Na^+^ and Ca^2+^, and outward repolarizing K^+^ currents [61,62]. The shape of the AP waveform is a reflection of the electrical function of the expressed ion channels (including their auxiliary subunits), and subtle changes in their well-regulated ion conductance (i.e., their time and/or voltage dependence) can have a substantial effect on AP morphology.

**Figure 1 ijms-22-02825-f001:**
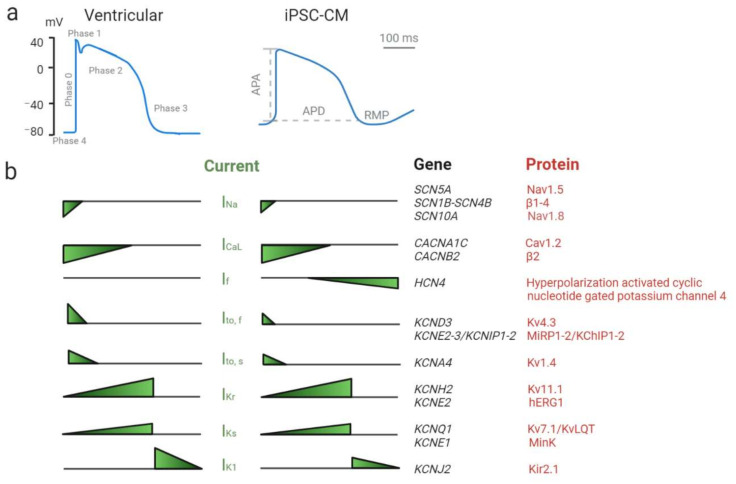
Ventricular action potential and underlying ionic currents (adapted from [32,61,63]). Schematic representation of human ventricular and human induced pluripotent stem cell-derived cardiomyocyte (iPSC-CM) action potential waveforms (**a**) with indicated in grey: action potential (AP) phases (on ventricular AP waveform) as well as action potential amplitude (APA), action potential duration (APD) and resting membrane potential (RMP) (on iPSC-CM AP waveform). Bottom panel represents the relative magnitude of the underlying ionic currents (**b**) together with the list of genes (in black) encoding alpha and auxiliary (beta) ion channel protein subunits (in red) which cooperate in generation of the represented ion currents (created with biorender.com December 2020).

Phase 0 of the ventricular AP, a rapid depolarizing upstroke, reflects the function of voltage-gated sodium channels (Na_v_1.5 with possible contribution of Na_v_1.8) generating the inward sodium current (I_Na_). It is followed by a transient repolarization of the cell membrane, phase 1, with inactivation of I_Na_ and activation of a transient outward voltage-gated K^+^ current (I_to_; consisting of a fast I_to,f_ and slower I_to,s_ component). This activity contributes to development of a characteristic prominent “notch” in the ventricular AP waveform. During the plateau phase, phase 2, the depolarization of the cell membrane activates voltage-gated Ca^2+^ channels, triggering the influx of Ca^2+^ (I_CaL_). While the Cav channels inactivate, the outward K^+^ currents dominate (I_Kr_ and I_Ks_) and drive further repolarization of the cell membrane in phase 3. Inwardly rectifying K^+^ channels (mediating I_K1_) contribute to late phase three repolarization and to the maintenance of the resting membrane potential (RMP) in phase four [61,62,64].

Unlike adult ventricular CMs, iPSC-derived CMs beat spontaneously due to a more positive RMP caused by reduced I_K1_ and presence of a pacemaker current (I_f_, generated by HCN4 channels). The depolarized state of the iPSC-CMs results in the inactivation of the majority of Na_v_1.5 channels, leaving very little I_Na_ available during phase 0 and reducing upstroke velocity, and leads to I_to_ inactivation, which reduces phase one repolarization. In addition, in iPSC-CMs I_to,s_ seems to be dominant compared to I_to,f_, while in adult CMs this is the other way round [61]. These are important characteristics to take into account when studying the electrophysiological phenotype of this model.

In the next paragraphs, we will briefly discuss different techniques employed in the investigations of BrS iPSC-CM electrophysiological parameters, including patch-clamping, calcium and voltage imaging, and micro electrode arrays (MEA).

### 3.1. Patch-Clamping

Considered the gold standard procedure in electrophysiological characterization, patch-clamping is a technique which allows for the measurement of ionic currents to study the behavior of the channels, or measurement of voltage changes across the cell membrane suited for AP characterization [63,65]. This technique allows for both the actual measurement of the membrane potential, enabling precise characterization of the AP, as well as the analysis of single ionic currents and ion channel activity. As this is an advantage when studying iPSC-CM disease models in detail, in all the discussed reports the patch-clamp technique has been applied (Table 1). Patch-clamping also offers the opportunity to resolve the more depolarized RMP of the iPSC-CMs and achieve a proper upstroke velocity and phase 1 repolarization by using dynamic clamp or injecting a sustained current to reach an RMP of −80–−90 mV. The disadvantages of the method remain its low throughput and its invasive and terminal nature, allowing only short-term recordings. Moreover, the need to record from single cells excludes the possibility to measure AP propagation between iPSC-CMs, though this clearly is a characteristic affected and worth studying in arrhythmias.

**Table 1 ijms-22-02825-t001:** Summary of the findings from published BrS iPSC-CM models. Table represents an overview of the identified gene variants, patient diagnosis, electrophysiological characterization methods used and obtained results. Data shown indicate the most important findings from investigated BrS iPSC-CMs models in comparison to control iPSC-CMs for action potential (AP) (APA—AP amplitude; APD—AP duration, dV/dT—upstroke velocity), sodium current (I_Na_) (current density, voltage dependence of activation/inactivation, recovery from inactivation), late sodium current (I_NaL_) (current density), calcium current (I_CaL_) (current density, activation/inactivation, recovery), potassium currents: I_to_, I_Ks_ and I_Kr_ (current density), calcium transients (CT) (CTD—CT duration, amplitude, rise rate, decay, presence of early afterdepolarization (EAD)/ delayed afterdepolarization (DAD)/arrhythmias) and field potential duration (FPD) characteristics. Legend: + performed analysis; − not performed analysis; ↑ significant increase/positive shift in voltage dependance; ↓ significant decrease/negative shift in voltage dependance; = no difference in comparison to control iPSC-CMs. BrS—Brugada syndrome; LQTS3/BrS—mixed Brugada syndrome and long-QT syndrome type 3 phenotype; Amp. —amplitude.

Reference	Gene	Variant	Diagnosis	EP Characterization Methods			Electrophysiological Findings																				
							AP			I_Na_				I_NaL_	I_CaL_				I_to_	I_Ks_	I_Kr_	CT					FPD
				Patch-Clamping	Ca^+^ Imaging	MEA	APA	APD	dV/dT	Current Density	Voltage Dependance of Activation	Voltage Dependance of Inactivation	Time of Recovery from Inactivation	Current Density	Current Density	Activation	Inactivation	Recovery	Current Density	Current Density	Current Density	CTD	Amp.	Rise Rate	Decay	EAD/ DAD/ Arrhythmias	
**Kosmidis et al. 2016** [66]	*SCN5A*	c.4912C>T (p.R1638X)	BrS	+	−	−	=	=	↓	↓	=	=															
		c.468G>A (p.W156X)	BrS				=	=	↓	↓	=	=															
**Liang et al. 2016 [67]**	*SCN5A*	c.2053G>A (p.R620H) and c.2626G>A (p.R811H)	BrS	+	+	−	=	=	↓	↓												↑	↓	↓		↑	
		c.4190delA (p.K1397Gfs*5)	BrS				=	=	↓	↓												=	↓	↓		↑	
**Ma et al. 2018 [68]**	*SCN5A*	c.677C>T (p.A226V) and c.4885C>T (p.R1629X)c.4859C>T	BrS	+	−	−	↓	↑	↓	↓	↓	↓	↑						=								
		p.T1620M	BrS				=	=	↑	=																	
**Selga et al. 2018 [69]**	*SCN5A*	c.1100G>A (p.R367H)	BrS	+	−	−				↓	↑	↓	↓														
**de la Roche et al. 2019 [56]**	*SCN5A*	c.2204C>T (p.A735V)	BrS	+	−	−	↓	=	↓	↓	↑	↓	=		=				↓								
**Davis et al. 2012 [70]**	*SCN5A*	c.5537insTGA (p.1795insD)	LQTS3/BrS	+	−	−	=	↑	↓	↓				↑													
**Okata et al. 2016 [71]**	*SCN5A*	c.5349G>A (p.E1784K)	LQTS3/BrS	+	−	+	=	↑	↓	=	=	↓		↑													↑
**El-Battrawy et al. 2019 [72]**	*SCN10A*	c.3749G>A (p.R1250Q) and c.3808G>A (p.R1268Q)	BrS	+	+	−	↓	=	↓	↓	=	=	↓	↓	↓	↑	↓		=	↓	=					↑	
**El-Battrawy et al. 2019 [73]**	*SCN1B*	c.629T>C (p.L210P) and c.637C>A (p.P213T)	BrS	+	+	−	↓	=	↓	↓	↑	↓	↑	↓	=	=	=	↑	=	↓	↓	=	=	=		↑	
**Belbachir et al. 2019 [74]**	*RRAD*	p.R211H	BrS	+	+	−	=	↑	↓	↓	=	=	↓	↑	↓							↑			↓	↑	
**Cerrone et al. 2014 [75]**	*PKP2*	c.1904G>A (R635Q)	BrS	+	−	−				↓																	
**Miller et al. 2017 [76]**	Undefined	Undefined	BrS	−	−	+																					=
	Undefined	Undefined	BrS																								=
	*PKP2*	c.302G>A (p.R101H)	BrS	+	−	+	=	↓	=																		=
**Veerman et al. 2016 [55]**	Undefined	Undefined	BrS	+	−	−	=	=	=	=	=	=	=														
	Undefined	Undefined	BrS				=	=	=	=	=	=	=		=				=								
	*CACNA1C*	int19 position -7 (benign)	BrS				=	=	=	=	=	=	=														

### 3.2. Calcium and Voltage Fluorescence Imaging

The use of calcium- or voltage- sensitive dyes in combination with fluorescence microscopy provides a non-invasive and high-throughput method for measuring intracellular Ca^2+^ fluctuations and membrane voltage changes in iPSC-CMs (single cells or growing in clusters or monolayers) [77,78]. With calcium imaging not only Ca^2+^ flux across the cell membrane but also intracellular Ca^2+^ oscillations defining the excitation–contraction coupling are captured, allowing more in-depth investigation of the intracellular Ca^2+^ balance. It should be noted though that iPSC-CMs show structural immaturity with a lack of T-tubules and reduced sarcomere organization, resulting in more immature calcium handling and excitation–contraction coupling. In four of the discussed reports [67,72,73,74] calcium imaging was used in addition to patch-clamping (Table 1). Nevertheless, the use of dyes prohibits long term recordings due to their phototoxicity and the signals have a high noise ratio [63,79].

To overcome these challenges, genetically encoded calcium and voltage indicators (GECI and GEVI) have recently been introduced [80]. The cells are then transfected with genes encoding fluorescent indicators, which consist of a Ca^2+^ or voltage sensing element coupled to one or two fluorophores that alter their fluorescent intensity based on conformational changes in the sensing element. An advantage of iPSC-CMs compared to native CMs in this case is that iPSC-lines with stable GEVI/GECI expression can be created, stored and repeatedly differentiated to cardiomyocytes. Several GECIs (GCaMP5G, R-GECO1) and one GEVI (ArcLight) have been used in control and patient-derived iPSC-CMs and shown to consistently represent the calcium transients [80,81,82,83,84] and the transmembrane APs [82,83,84,85,86]. Although these reporters offer significant advantages over the traditional dyes, a major drawback is that the transgenes integrate randomly into the genome, raising serious concerns about potential gene disruption and alteration of local and global gene expression that could adversely affect normal cellular functions. Thus far, these genetically encoded indicators have not been used in BrS disease modelling yet.

### 3.3. Micro-Electrode Array (MEA)

MEA platforms offer the possibility of high-throughput, non-invasive, label-free and long-term measurement of extracellular field potentials (FPs) from clusters and monolayers of iPSC-CMs [63,87,88,89,90,91]. It requires the iPSC-CMs to be seeded on MEA plates, with multiple microelectrodes embedded in 2D grids in the cell culture surface. MEAs have the advantage that AP conduction velocity over a layer of iPSC-CMs can be measured. Conventional arrays have a limited spatial resolution (typically 100 µm) due to the limited number of electrodes in the grid. A new class of MEAs that is based on complementary metal–oxide semiconductor (CMOS) technology has been developed as a solution to this limitation [89,92]. In CMOS-MEAs, thousands of microelectrodes are arranged at high spatial resolution on a chip, tremendously increasing the amount of information that can be gathered from a single iPSC-CM culture. In only two of the discussed articles MEA was used to study BrS iPSC-CMs [71,76] (Table 1), in one to confirm field potential duration (FPD) prolongation correlating to the long QT syndrome (LQTS) part of a mixed phenotype [71] and in the other to perform drug challenges on the cells [76].

## 4. Findings from Published BrS iPSC-CM Models

Most of the published functional reports from iPSC-CM models, focused on sodium channel genes such as *SCN5A* [56,66,67,68,69,70,71], *SCN10A* [72] or *SCN1B* [73]. In addition, pathogenic variants in other BrS-related genes, such as *RRAD* [74] or *PKP2* [75,76] were modelled. Finally, as BrS is a complex disease with a large proportion of cases being genetically unresolved, characterization of iPSC-CM models from patients with undefined genetic cause has also been performed [55,76]. In this review, we will summarize the findings from these thirteen reports, with a focus on the electrophysiological results. An overview of the studied genes, exact investigated genetic variants and electrophysiological results obtained is given in Table 1 and extra information is available in the Appendix A (used reprogramming and differentiation approach with additional patient information are summarized in Appendix A; reported electrophysiological parameters for AP, CT, FPD and tested ion channels are summarized in Appendix A).

### 4.1. Sodium Channel Genes

#### 4.1.1. Sodium Channel α-Subunits—SCN5A

BrS is mainly associated with loss-of-function mutations in *SCN5A*, leading to reduced I_Na_ due to lower expression levels or the production of defective Na_v_1.5 protein, while gain-of-function mutations in *SCN5A* contribute to LQT syndrome 3 (LQTS3) [93,94]. In some cases, an overlap of the symptoms of both arrhythmias can be observed in patients carrying specific *SCN5A* mutations [95,96,97]. Until now, ten iPSC-CM models of BrS-related *SCN5A* variants have been published. Eight of them were generated from patients with a pure BrS phenotype [56,66,67,68,69] and two from patients with a mixed LQTS/BrS phenotype [70,71] and will be discussed as such in the following paragraphs.

##### Pure BrS Phenotype

Kosmidis et al. [66] generated two iPSC-CM models with nonsense *SCN5A* variants, for which they recruited one BrS patient with a more severe phenotype carrying a p.(Arg1638*) variant and one with a relatively mild phenotype related to a p.(Trp156*) variant (Appendix A). Both variants caused a reduction in I_Na_ density and slowed down the upstroke velocity of the AP in the iPSC-CMs (Appendix A) compared to iPSC-CMs from two unrelated control individuals. There were no significant differences in the activation or inactivation process of the I_Na_ between patient iPSC-CMs and controls. As both modelled variants were predicted to generate a premature stop codon, the presence of nonsense mediated decay (NMD) was investigated in cloning experiments. 61% of patient iPSC-CMs expressed the p.(Arg1638*) variant, confirming that its location in the last exon caused NMD escape, while only 19% of patient iPSC-CMs expressed the p.(Trp156*) variant, confirming the occurrence of NMD. The authors tested two nonsense readthrough-promoting drugs, gentamycin and PTC124, on patient iPSC-CMs to investigate their therapeutic potential on the nonsense variants. Unfortunately, they did not observe any significant increase in I_Na_ in treated iPSC-CMs in comparison to baseline conditions, leading to their conclusion that these drugs are unlikely to represent an effective treatment for patients carrying the studied mutations.

Liang et al. performed the characterization of two BrS cases: one (BrS1) with double missense *SCN5A* mutation: p.(Arg620His) and p.(Arg811His), and a second one (BrS2) with a c.4190delA; p.(Lys1397Glyfs*5) frameshift *SCN5A* mutation [67]. For comparison, iPSC-CMs were also created of two healthy unrelated control individuals. Both patient iPSC-CMs showed AP profiles of a closely coupled single triggered beat (in 39.6% and 34.5% of recordings in BrS1 and BrS2, respectively), and of sustained triggered activity (5.6% and 6.8% in Brs1 and BrS2, respectively), increased peak-to-peak interval variability and slower upstroke velocity (Appendix A). Sodium current analysis in both patient iPSC-CMs showed visible I_Na_ reduction (Appendix A), which correlated with a lower protein expression of Na_v_1.5 in comparison to controls. Calcium imaging experiments showed about 60% reduction in Ca^2+^ transient amplitude, 50–80% decreased rise rate and increased variation in beating intervals in both patient iPSC-CMs (Appendix A). RNA-seq revealed a closer homology in overall gene expression profile between the two BrS iPSC-CMs compared to the two control lines, suggesting disease-specific gene expression changes. RT-qPCR analysis confirmed a reduced expression of *SCN5A*, *KCND3* and *KCNJ2* in patients compared to control iPSC-CMs, suggesting reduced Ito and reduced I_K1_ could also be involved in the arrhythmic phenotype. Finally, they used the CRISPR/Cas9 technique to correct the c.4190delA variant in BrS2 patient iPSCs. Genome edited iPSC-CMs (BrSp2-GE) showed a marked reduction in arrhythmic activity, improved upstroke velocity (Appendix A) normalized calcium transient (CT) parameters (Appendix A), increased Na_v_1.5 membrane expression and a partial rescue of the I_Na_ density (Appendix A). As such, the authors were able to prove that the frameshift variant was solely responsible for the observed cellular phenotype.

Ma and colleagues reported an iPSC-CM model from a compound heterozygous *SCN5A* mutation carrier, carrying p.(Ala226Val) and p.(Arg1629*) (BrS1) and a healthy sibling control (CON1, his unaffected brother) [68]. They also created iPSC-CMs with a milder p.(Thr1620Met) mutation through genome editing (BrS2) and used an extra control iPSC-CM line of commercial iCell^®^ Cardiomyocytes (Fujifilm Cellular Dynamics, Inc., Madison, WI, USA) (CON2). BrS1 iPSC-CMs showed 50% reduced SCN5A mRNA expression in comparison to CON1 iPSC-CMs, suggesting the occurrence of NMD and corresponding with an observed 75% reduction in I_Na_ (Appendix A). At first, the AP recordings of patient iPSC-CMs did not show significant differences in comparison to controls, which may be explained by the more positive RMP of these cells (about −45 mV) at which most of the I_Na_ is not available and I_to_ is inactivated (Appendix A). The application of an in silico I_K1_ injection revealed a >75% reduction in upstroke velocity and a reduction in AP amplitude in BrS1 iPSc-CMs (Appendix A) and appearance of phase 1 repolarization, only in CON1 iPSC-CMs. Moderate changes in the rate of steady-state activation and inactivation and a more significant change in the rate of recovery from the inactivation of the sodium channels was observed in BrS1 compared to CON1. BrS2 iPSC-CM characterization revealed normal I_Na_ and normal APs, with upstroke velocity (dV/dT) max even higher than in both controls (Appendix A). Since BrS pathology is linked with slow heart rates, AP recordings at different lower pacing frequencies were performed. In 25% of BrS1 patient cells, at 0.1 Hz an average 66% action potential duration (APD)90 reduction was observed, while APD prolongation or moderate shortening was observed in the rest of the patient iPSC-CMs and the control cells. (Appendix A). The observed marked reduction in APD, represented an increased phase 1 repolarization and loss of phase2 AP pattern, highly resembling the proarrhythmic loss-of-dome in epicardial ventricular CMs, fitting with the repolarization disorder hypothesis. I_to_ measurements in patient and control iPSC-CMs showed similar current density with an increase in the low pacing rate. To investigate I_to_ influence on the observed phenotype in BrS1 iPSC-CMs, the cells were treated with 4-aminopyridine (4-AP, blocker of I_tof_ and I_tos_) and APs were measured at 0.1 Hz. The authors noticed that 0.5 mM 4-AP completely reversed the increased phase one repolarization and loss of phase two dome (Appendix A). They concluded that I_Na_ and I_to_ could play a coordinated role in BrS causation, where loss of I_Na_ together with heterogeneous elevated I_to_ in a fraction of iPSC-CMs (at lower heart rates) make the ventricular CMs undergo proarrhythmic changes. Such an observation would not have been possible in a heterologous expression system and supports the value of iPSC-CMs to reveal interplay between different currents in CMs.

Selga et al. performed a comparison of I_Na_ properties in iPSC-CMs from a BrS patient carrying a p.(Arg367His) *SCN5A* variant and an unrelated healthy control individual (two clones each), obtained using two differentiation protocols: an EB-based spontaneous differentiation and monolayer-based differentiation (Appendix A) [69]. The authors reported similar observations from both differentiation protocols: a similar reduction in peak I_Na_, a positive shift in the voltage dependence of channel activation (i.e., channel opening) concomitantly with a negative shift in voltage dependence of inactivation (a process that makes the channels non-conductive upon opening), reflecting a clear loss of function (Appendix A). They also reported accelerated recovery from inactivation in patient iPSC-CMs in comparison to the control in both tested groups. Interestingly, experiments in tSA cells revealed only the I_Na_ reduction of about 48% for the mutant protein, while the steady-state activation/inactivation properties were not changed in comparison to the wild type (WT). This again shows an added value of the iPSC-CM model, revealing pro-arrhythmic channel function changes that were not detected in a conventional heterologous expression system, probably due to the absence of auxiliary subunits.

In 2019, de la Roche and colleagues used CRISPR/Cas9 technology to introduce a homozygous p.(Ala735Val) *SCN5A* variant in a healthy control iPSC line and obtained two independently derived mutant clones (MUT1 and MUT2) [56]. They state that they used both isogenic and non-genetically related iPSC-CM controls (“wild type” WT) for the electrophysiological characterization. Long-term cultivation (27–42 days) of the iPSC-CMs on a stiff matrix was applied to promote the maturation of the cells. AP recordings carried out with a hyperpolarizing current (~I_K1_) injection showed similar APD50 values in WT and A735V iPSC-CMs, but significantly reduced action potential amplitude (APA), 74% reduction in upstroke velocity and A735V cells not displaying a sharp AP peak with phase one repolarization notch (Appendix A). These recordings strongly indicated altered activation or inactivation characteristics of the mutant Na_v_1.5 channel. They measured I_CaL_ contribution to the I_Na_ current and showed the same I_CaL_ density in both groups (Appendix A). A735V iPSC-CMs showed a +30 mV depolarizing shift in the voltage dependence of channel activation and a negative shift in the voltage dependence of inactivation, resulting in the reduced availability of the channels for AP generation at physiological membrane potentials (60% of inactivated channels in mutant and 40% in WT at −80mV) (Appendix A). In simultaneous experiments in HEK cells, the reduction in I_Na_ as well as the shift in channel activation were observed, however, no differences in the voltage dependence of the inactivation of the mutant channel were noted. This again confirms that different cellular composition can lead to different channel characteristics and emphasizes the importance of the iPSC-CM model. De la Roche et al. investigated differences between two mutant and two WT clones and between five independent differentiation experiments in both groups. As they did not see significant differences in values between clones and differentiations, they were able to pool the data together for the comparisons (Appendix A).

##### Mixed Phenotype (BrS/LQTS)

Davis et al. used iPSC-CMs to model an *SCN5A* p.(1795insAsp) mutation, identified in a large family where mutation carriers presented with variable phenotypes, including diagnosed cases of BrS as well as LQTS and cardiac conduction defects (Appendix A) [70]. This mutation was previously modelled in cardiomyocytes isolated from transgenic mice, where a prolongation of APD was observed together with a slowing of the upstroke velocity and reduced I_Na_ density but unchanged kinetic properties of this current [98,99]. However, experiments in expression systems (HEK 293) showed a disruption of the fast inactivation, causing a sustained Na^+^ current (i.e., a late sodium current, I_NaL_) throughout the action potential plateau and prolonging cardiac repolarization at slow heart rates (explaining LQTS phenotype), as well as an increase/augmentation in the slow inactivation component, delaying the recovery of the sodium channel availability between stimuli and reducing the I_Na_ at rapid heart rates (explaining BrS phenotype) [100]. In the present study, they first used mouse iPSC-CMs generated from heterozygous mutant and wild type mice and showed reduced upstroke velocity, prolonged APD and I_Na_ reduction in the mutant miPSC-CMs but no changes in the voltage dependence of the activation and inactivation parameters. In their human iPSC-CMs, Davis et al. observed a 54% reduction in I_Na_ density and an increase in I_NaL_ (Appendix A) compared to control iPSC-CMs from an independent healthy individual, but they did not characterize activation or inactivation kinetics. Action potential analysis revealed significantly reduced upstroke velocity and prolonged APD90, recapitulating both arrhythmia phenotypes (Appendix A).

Another paper reporting an iPSC-CM model of a p.(Glu1784Lys) *SCN5A* mutation, underlying/causing a mixed BrS/LQTS phenotype, was published by Okata et al. [71]. They used control iPSC-CMs from two unrelated healthy individuals and reported FPD (MEA) as well as APD90 (patch-clamp) prolongation in patient iPSC-CMs (Appendix A). No differences in upstroke velocity as well as I_Na_ density were observed in patient iPSC-CMs, however, they observed an increase in I_NaL_ in patient iPSC-CMs (Appendix A). As they observed a high expression of *SCN3B* in iPSC-CMs compared to adult CMs and in experiments with tsA-201 cells, they saw that SCN3B co-expression with the mutant E1784K-SCN5A protein increased peak I_Na_ and caused a positive shift in the voltage dependence of channel inactivation compared to WT. They performed SCN3B siRNA knockdown in patient iPSC-CMs to check if the expression of this embryonic type Na^+^ channel β-subunit can mask the BrS phenotype. In iPSC-CMs with SCN3B knockdown, they reported a decrease in I_Na_ density and a negative shift in the voltage dependence of channel inactivation only in the patient iPSC-CMs, explaining the BrS phenotype related to the *SCN5A* mutation. To investigate if the *SCN5A* p.(Glu1784Lys) mutation is sufficient to produce the observed electrophysiological changes, they generated an isogenic control of the patient iPSCs, using an adenoviral vector. Corrected iPSC-CMs showed shortened APD90 (Appendix A), no significant difference in I_Na_ density but a significant decrease in I_NaL_ (Appendix A) compared to the patient iPSC-CMs. They did not discuss this, but it seems they did not perform SCN3B knockdown in these experiments, which could explain these sodium current results. Those results confirmed that the p.(Glu1784Lys) *SCN5A* mutation contributes solely to the development of the mixed LQTS/BrS phenotype, and that the *SCN3B* expression in iPSC-CMs can mask the electrophysiological properties characteristic to BrS syndrome.

#### 4.1.2. Sodium Channel α-Subunits—SCN10A

Genome-wide association study (GWAS) analysis of QRS interval duration led to the first identification of *SCN10A* as a candidate gene in cardiac ventricular conduction disorders [15]. In addition, a GWAS for BrS also detected an association with common variants in *SCN10A* (coding for the Nav1.8 channel) [19]. These studies were followed by multiple reports of *SCN10A* variants involved in development of BrS, together with their functional analysis in expression systems, suggesting *SCN10A* is an important player in BrS etiology [101,102,103,104,105]. Recently, El-Battrawy et al. reported a successful attempt of the iPSC-CM modelling of a double *SCN10A* mutation (p.(Arg1250Gln) and p.(Arg1268Gln)) related to the BrS phenotype [72]. They used three clones of the patient iPSCs, as well as iPSC-CMs from three independent healthy control individuals. They observed reduced peak I_Na_, reduced I_NaL_ and accelerated recovery from inactivation in patient iPSC-CMs (Appendix A), contrasting with the increased expression of both *SCN5A* as well as *SCN10A* mRNA in patient iPSC-CMs in comparison to controls. El-Battrawy and colleagues also detected about a 50% reduction in *KCNJ2* expression in patient iPSC-CMs in comparison to controls, without changes in I_K1_. Regarding calcium and other potassium currents, they observed reduced peak density, a positive and negative shift of the voltage dependence of channel activation and inactivation, respectively, of I_CaL_ (Appendix A), together with a reduction in I_Ks_ (Appendix A). AP characterization showed reduced APA and upstroke velocity, but similar APD (Appendix A). An ajmaline addition revealed an increased susceptibility of patient iPSC-CMs to the sodium blocker, visible in reduction in APA as well as upstroke velocity already with a 3µM ajmaline addition (Appendix A). They investigated the presence of arrhythmic events in patient iPSC-CMs using calcium imaging, which revealed an increased frequency of EAD-like events in patient iPSC-CMs (90%) in comparison to controls (50% and 45% in two control lines).

#### 4.1.3. Sodium Channel β-Subunits—SCN1B

The proper function of the cardiac Na_v_1.5 channel is known to be regulated by the β-subunits of the channel complex [106]. It was previously reported that variants in *SCN1B*, encoding the β1-subunit, associate with BrS, including successful investigation in expression systems [107]. El-Battrawy et al. performed electrophysiological investigations of compound variants in *SCN1B* (p.(Leu210Pro) and p.(Pro213Thr)) in patient iPSC-CMs compared with control iPSC-CMs of three independent healthy individuals [73]. They reported increased expression levels of *SCN5A* and decreased expression levels of *CACNA1C*, *KCNJ2*, *KCNH2*, *SCN1B* and *SCN3B* in the patient iPSC-CMs, while on the protein level, they observed reduced *SCN1B* and similar Na_v_1.5 expression in comparison to controls. Electrophysiological investigation showed reduced peak I_Na_ and I_NaL_ together with a positive shift in the voltage dependence of activation and negative shift of the inactivation process as well as decelerated recovery from inactivation of I_Na_ in the patient iPSC-CMs (Appendix A). When characterizing other ion channels, El-Battrawy et al. reported decelerated recovery from inactivation for I_CaL_, as well as a reduction in I_Ks_ and I_Kr_ and no change in I_to_ in patient iPSC-CMs (Appendix A). They postulated that these changes probably resulted from secondary changes induced by the *SCN1B* variants. AP analysis of patient iPSC-CMs revealed reduced APA and upstroke velocity, while APD did not change (Appendix A). They investigated the effect of ajmaline addition (30 µM) and observed a more pronounced change in APA as well as upstroke velocity in patient iPSC-CMs in comparison to controls, especially at higher beating frequencies, indicating a higher sensibility of BrS cells to ajmaline application. Using calcium imaging, they observed no differences in Ca^2+^ transients, but increased arrhythmia like events in patient compared to control iPSC-CMs (85% vs. 45% of cells). After treatment with 10 µM of carbachol, a parasympathetic stimulator, control cells showed a reduction in the beating frequency, whereas the patient iPSC-CMs showed an increase in the beating frequency, an observation that they could not explain.

### 4.2. iPSC-CM Models of Variants in Other Arrhythmia-Related Genes

#### 4.2.1. Calcium Channel Related Proteins

Whole exome sequencing analysis has been proven a powerful tool in genetic screening of BrS patients. WES analysis of five familial BrS patients performed by Belbachir et al. led to the identification of a p.(Arg211His) variant in the *RRAD* gene [74]. *RRAD* encodes RAD (Ras associated with diabetes) GTPase, known to play a role in Ca_v_1.2 trafficking and associated with ventricular arrhythmia in mice. The variant identified by Belbachir and colleagues has a Combined Annotation Dependent Depletion (CADD) score of 33, occurs only once in GnomAD and co-segregated with disease phenotype in the patient family, indicating potential pathogenicity of the variant, and their studies in mouse CMs suggested a gain-of-function effect. Sanger sequencing of *RRAD* coding regions in 186 unrelated BrS patients led to the identification of three additional rare mutations, showing a trend for *RRAD* variant enrichment in BrS patients in comparison to 856 tested control individuals. The authors derived iPSC-CMs of two patients (BrS1 and BrS2) and two unaffected non-carrier siblings (Ctl1 and Ctl2) and investigated both I_Na_ and I_CaL_ properties, together with AP and CT analysis (Table 1). Electrophysiological tests on patient iPSC-CMs showed the presence of slower spontaneous rhythms, prolonged AP duration and reduced upstroke velocity in comparison to Ctl1 (Appendix A). BrS1 iPSC-CMs also showed a ~40% reduction in I_Na_, with a slight acceleration of recovery from inactivation in comparison to Ctl1 (Appendix A). The authors correlated these changes with a lower expression of Na_v_1.5 protein in patient compared to control iPSC-CMs, while on the transcript levels there was no detectable difference. They reported a larger persistent Na+ current I_NaL_ and ~30% reduced I_CaL_ in BrS1 iPSC-CMs (Appendix A). CT results showed the presence of early afterdepolarizations (EADs), slower calcium transient decay and calcium transient duration (CTD) prolongation in patient iPSC-CMs (Appendix A). To prove the pathogenicity of the *RRAD* variant, CRISPR/Cas9 was used to knock it in in an unrelated control iPSC line (Rad R211H ins in Appendix A). Similar to patient iPSC-CMs, genome-edited cells showed slowed spontaneous rhythms, lower upstroke velocity, prolonged APD, reduced peak I_Na_ and increased persistent Na^+^ current compared to the isogenic WT control (Appendix A). No I_CaL_ differences were observed between the lines, suggesting no significant effect of the variant on Ca_v_2.1 trafficking. However, calcium imaging showed an uneven beat rate, with EADs in 20% of recorded cells and slowed calcium reuptake in the genome-edited iPSC-CMs (Appendix A). Using immunostaining and confocal microscopy, Belbachir et al. observed cytoskeleton defects with impaired F-actin organization and cortical distribution of troponin I in 70% of patient iPSC-CMs. This led to a decrease in cell contractility and focal adhesion formation and patient cells showed a preferential round shape, with increased thickness in comparison to flat control iPSC-CMs [74]. Genome edited knock-in iPSC-CMs confirmed the influence of RRAD p.(Arg211His) on the observed phenotype, as the cytoskeletal defects were now detected in ~40% of the tested cells. The authors concluded that due to preferred expression of RAD in the RVOT, the decreased cell–cell connection could lead to abnormal cardiac conduction in that region and cause BrS. All these results confirmed a causal role of the p.(Arg211His) *RRAD* variant in development of the BrS phenotype.

#### 4.2.2. Desmosomal Proteins

Although desmosomal protein abnormalities are predominantly linked with arrhythmogenic (right ventricular) cardiomyopathy (ARVC or ACM) [108,109], several studies showing the association of variants in *PKP2* with the BrS phenotype have been reported [75,110]. Two BrS-related PKP2 variants of unknown significance have so far been modelled in iPSC-CMs [75,76]. Although Cerrone et al. performed only a small part of their study in iPSC-CMs, they showed that the p.(Arg635Gln) variant in *PKP2* alone can contribute to a significant reduction in peak I_Na_ density [75]. As their experiments in HL1 cells showed that knockdown of the endogenous *PKP2* of the cells in combination with co-expression of the p.(Arg635Gln) variant (as well as other BrS-related missense variants) with WT PKP2 led to a reduction in peak I_Na_, they performed a rescue experiment in the iPSC-CM line derived from an ARVC patient with a homozygous c.2484C > T *PKP2* frameshift loss-of-function variant [111] using lentiviral constructs containing WT-PKP2 and PKP2-R635Q. At baseline, I_Na_ was significantly reduced in the patient iPSC-CMs compared to the control embryonic stem cell (ESC)-CMs obtained from H9 ESCs (Appendix A) and the rescue experiments showed that only WT-PKP2 transduction led to a significant increase in I_Na_ density. These results confirmed that I_Na_ depends on the expression and structural integrity of *PKP2*.

A p.(Arg101His) variant in *PKP2* (classified as VUS or likely benign) was present in one of the patient iPSC-CMs investigated by Miller et al. in a study aiming to identify ajmaline’s mode of action in iPSC-CMs and test whether differences in ajmaline response could be determined between BrS patients and controls on a MEA platform [76]. AP analysis revealed no differences in upstroke velocity, while APD90 was significantly reduced in patient iPSC-CMs in comparison to the control (Appendix A). Although a prolongation of the FPD in the p.(Arg101His) patient iPSC-CMs was visible in the presence of 100 nM of ajmaline (average 43.8 ms; 1.1-fold) in comparison to baseline conditions, while this was not observed in iPSC-CMs from an unrelated healthy control individual (iPSC-HS1M; Appendix A), the authors noted comparable FPD prolongation in the presence of 1, 10 and 100 µM of ajmaline in patient and control iPSC-CMs (Appendix A). They concluded there was no significantly increased susceptibility to ajmaline in the patient cell lines (see also further below). The authors provided proof of ajmaline blocking both I_Na_ and I_Kr_ on control iPSC-CMs, however, in the scope of this study they did not include the characterization of any of the underlying currents of the patient iPSC-CMs, but did show reduced APD90 using patch-clamp [76].

### 4.3. Unknown Genetic Contribution

Attempts have been made in the iPSC-CM modelling of BrS phenotypes from patients with unknown genetic causes [55,76]. In the study by Veerman et al., iPSC-CMs were generated from three BrS patients with spontaneous BrS type-1 ECG pattern. Two did not carry variants in *SCN5A* and other BrS-related genes, while one patient carried a *CACNA1C* variant in intron 19 (position −7) that was predicted not to affect the splicing of the gene and was classified as benign. For the electrophysiological characterization, they used iPSC-CMS from two unrelated healthy control subjects (iCtrl1 and iCtrl2) as well as a positive iPSC-CM control (*SCN5A* p.(1795insAsp) carrier; iSCN5A Appendix A). While in this positive control, Veerman and colleagues observed APD prolongation and the expected dV/dT max reduction, in the analyzed patient lines they did not see significant differences in all tested AP properties in comparison to the negative controls (Appendix A), although in one patient line APD was significantly shorter compared to one of the control lines (iCtrl2). Similarly, they did not detect any differences in I_Na_ density in patient iPSC-CMs in comparison to negative controls, while in the positive control, they saw a significant reduction in the peak current (Appendix A). Interestingly, the authors observed significant differences in V_1/2_ voltage dependence of the inactivation of I_Na_ and I_CaL_ between iCtrl1 and iCtrl2 and the investigated patient iPSC-CMs were similar to one or the other control cell line (Appendix A), showing variability in channel inactivation kinetics in iPSC-CMs. Nevertheless, in summary, the authors were not able to observe a BrS phenotype in their iPSC-CMs obtained from BrS patients with unknown genetic contributions.

In the study by Miller et al., investigating the effects of ajmaline, two out of three selected iPSC-CM lines were obtained from BrS patients with unknown genetic causes and spontaneous BrS type-1 ECG pattern. The authors observed consistent FPD prolongation in all of the tested patient clones (iBR1-P5M-L1, iBR1-P5M-L9 and iBR1-P6M-L1) with addition of increasing ajmaline concentrations (100 nM–100 µM) with a maximum 1.46-fold increase at a 100 µM ajmaline concentration (Appendix A). However, no significant differences in ajmaline susceptibility could be observed between patient and control iPSC-CMs [76]. This study did demonstrate that iPSC-CMs were suitable to test the blocking effect of ajmaline on both depolarization (I_Na_) and repolarization (I_Kr_).

## 5. Conclusions and Future Perspective

A decade of disease modelling using the novel iPSC-CM approach in the field of cardiac arrhythmias and Brugada syndrome has proven its feasibility in modelling patient-specific cellular phenotypes (Figure 2). In this review, we gathered information from thirteen currently available reports studying BrS-specific iPSC-CMs and discussed the resulting electrophysiological findings.

These findings confirmed that the main electrophysiological changes in BrS iPSC-CMs affect sodium channel activity, as previously observed in expression systems and murine models. In particular, a reduced peak I_Na_ density [56,66,67,68,69,70,72,73,74,75] was observed in all the described patient iPSC-CMs with an identified mutation, both in *SCN5A* or the other genes. So, all genetic variants led to the development of a phenotype with reduced I_Na_. (Table 1 and Appendix A). When sodium channel kinetics were studied, mostly a negative shift in the voltage dependence of channel inactivation (leading to a higher number of non-conductive inactivated channels) [56,68,69,71,73], a positive shift in the voltage dependence of channel activation (i.e., stronger membrane depolarization is required to open the channels) [56,69,73] and in two reports, an increased time of recovery from inactivation (leading to the prolonged inactivation of the channels) [68,73] was observed, all fitting with the reduced I_Na_ phenotype. In one report, a negative shift in the voltage dependence of channel activation [68] and in three reports, an accelerated recovery from inactivation [69,72,74] were observed, which were not explained with regard to the phenotype. Interestingly, in two papers [56,69], changes in channel kinetics were not detected in a conventional heterologous expression system, confirming the value of the iPSC-CM model expressing a more complete CM intracellular environment, including channel auxiliary subunits. In line with the sodium current defects, reduced upstroke velocity was observed in 11 out of 15 models, where AP properties were tested [56,66,67,68,70,71,72,73,74] and APA reduction was seen in four models [56,68,72,73] (Table 1 and Appendix A). When other currents apart from I_Na_ were investigated, mostly when modelling variants in other genes than *SCN5A*, also I_CaL_ [72,73,74] and/or in some instances I_Ks_ [72,73], I_Kr_ [73] as well as I_to_ [56] property changes were reported. Taking into account the oligogenic nature of BrS, it would be interesting to study several currents in BrS iPSC-CM models.

Interestingly, in two reports authors noticed microstructural changes in their disease models, which could contribute to the observed electrophysiological abnormalities [74,75]. Belbachir and colleagues observed that the *RRAD* p.(Arg211His) variant disturbed F-actin organization in patient iPSC-CMs, compromising focal adhesion and potentially cell–cell communication and AP conductance, mainly in the RVOT where RAD is expressed [74]. Cerrone et al. reported a microtubule abnormality in adult CMs from a heterozygous *PKP2* p.R635Q mouse, they did not study this in their iPSC-CM model. As sodium channel proteins are transported via the microtubule network, their results show that *PKP2* deficiency can affect the ability of microtubules to reach intercalated discs and impair Nav1.5 expression there [75]. It has already been suggested that disturbances in *PKP2*, and other desmosomal or junctional proteins at the CM intercalated discs (the “connexome”) can affect the trafficking or activity of sodium channels over there [91,112,113,114] explaining the link between structural abnormalities, electrophysiological changes and BrS. The interplay and dependency of all these intercalated disc proteins, integrating both mechanical and electrical functions of the CMs, have been beautifully illustrated in recent reviews [115,116,117]. In addition, in clinical studies, discrete structural abnormalities in the right ventricle of BrS patients have been observed [23,118,119]. Based on these insights, it would be interesting to look in more detail into structural changes in all BrS iPSC-CM models generated, since these could be part of the primary defect or play a modifying role in the disease phenotype.

iPSC-CMs from carriers of *SCN5A* variants presenting with a mixed BrS/LQTS pathology were shown to recapitulate both phenotypes: reduced upstroke velocity (i.e., dV/dT max) or increased I_NaL_ as well as APD prolongation was noted [70,71] (Table 1). Interestingly, APD changes were also observed in iPSC-CMs from pure BrS patients, one carrying a compound heterozygous *SCN5A* mutation (p.(Ala226Val) and p.(Arg1629*)) and one a p.(Arg211His) *RRAD* variant, where APD prolongation was noted [68,74], and one with a *PKP2* p.(Arg210His) variant, where APD was reduced in comparison to control [76]. Similarly, in studies where Ca^2+^ imaging and MEA technology were used, calcium transient and FPD differences have been reported in patient iPSC-CMs in comparison to controls (in six out of seven tested BrS iPSC-CMs at baseline) (Appendix A) [67,71,74].

An interesting advantage of iPSC-CM models is that they provide the opportunity to study cellular disease mechanisms in patients without an identified mutation, and two groups indeed investigated this. In one study [55], iPSC-CMs of three different patients showed no changes in sodium currents, nor other AP characteristics studied, even at different pacing frequencies. In the other study [76], only the effect of ajmaline on FPD was investigated and no increased susceptibility to ajmaline was observed in the two studied patients. Perhaps very subtle or non-studied electrophysiological changes were present in the models. Another explanation could be that the clinical phenotype was not caused by defects in ion currents but involved structural abnormalities at the cell and/or tissue level or disturbances of other unexpected pathways. This again supports the importance of in-depth study of the generated iPSC-CM BrS models, and the value of such models containing the full CM intracellular protein repertoire as well as the possibility to create and study more tissue-like structures. When the genetic cause is unknown or when modifiers underlying variable disease expressivity are sought, an approach including transcriptome and proteome analysis of iPSC-CMs could provide a more holistic picture and valuable indications on involved genes, proteins or pathways.

The fact that patient iPSC-CMs carry the patient’s full genetic background, including potential modifiers, can be a drawback of the model as well, when trying to establish the causal relationship between genotype and phenotype and to prove the pathogenic effect of a specific genetic variant. This is especially important when evaluating VUS or potential novel BrS candidate genes. In such cases, genetic engineering, such as CRISPR Cas genome editing, should be applied to correct the studied variant in the patient iPSCs (creating an isogenic control) or introduce it into one or more control iPSC-lines. The study of these models will reveal whether the investigated variant is necessary and sufficient to cause the phenotype or suggest the contribution of other genetic factors in the observed phenotype. In four of the discussed reports, genetic engineering was used to create isogenic controls [67,71] or introduce the variant of interest on a control background [56,74]. In each study, the modified iPSC-CMs showed rescue or development of the phenotype [56,67,71,74], confirming the causative status of novel variants and in one case, providing evidence for the involvement of SCN3B in subtle changes in the BrS/LQTS phenotype [71].

Over the years, it has been shown that iPSC-CMs display an immature phenotype with an expression pattern resembling that of fetal human cardiomyocytes [31,32,120,121]. They are spontaneously active (like fetal hCMs) due to a substantially reduced I_K1_ density and the presence of a pacemaker current. Though sodium channel expression is normal in iPSC-CMs, the more positive RMP results in the inactivation of the channels, leaving very limited I_Na_ available during the phase 0 depolarization and thereby potentially masking the impact of I_Na_ deficiency on the AP, more specifically the upstroke velocity. The depolarized state of iPSC-CMs could also lead to I_to_ inactivation and the absence of phase one repolarization in iPSC-CMs. This obviously poses a problem for BrS modelling, leading to the choice of several groups to impose I_K1_ onto their iPSC-CMs to achieve an RMP of −80–−90 mV and a more mature electrophysiological phenotype. It should be noted that this is performed on isolated cells, deprived of cell–cell contacts that are also potentially important for current expression, such as I_Na_. It is clear from Appendix A that in studies without I_K1_ injection and with depolarized RMP of the iPSC-CMs, the values of upstroke velocity were much lower than the normal range of native adult CMs and it is questionable whether they represent trustworthy values and true differences between patient and control models. Other interesting observations brought forward in the discussed articles, are the likely different expression profiles of I_to,f_ and I_to,s_ in hiPSC-CMs compared with that in adult epicardial ventricular CMs (I_to,s_ dominant in the first and I_to,f_ dominant in the latter) [61], and the expression of a foetal form of sodium channel β-subunit (*SCN3B*) that could be masking the BrS phenotype by compensating I_Na_ reduction [71]. In addition, iPSC-CMs show structural immaturity including a lack of T-tubules, and reduced sarcomere organization. This has been shown to affect calcium handling and excitation–contraction coupling in iPSC-CMs [32]. Additionally, in light of the previously mentioned potential microstructural abnormalities in BrS, this could affect read-outs and should certainly be taken into consideration when interpreting results. Methods to mature iPSC-CMs, such as hormone treatments, mechanical and electrical stimulation, 3D cell culture etc., are hot topic of investigations and are anticipated to show their value in future studies. It should be noted as well that every model has it limits, but it is important to recognize them and take them into account when interpreting and extrapolating data.

Since the differentiation of iPSCs to CMs is not fully defined and the addition of Wnt signaling pathway influencing factors is certainly a minimization of all processes present during actual embryonic development, the current methodology results in a heterogeneous iPSC-CM culture (mixed with non-CMs) that can vary with every differentiation round. To create robust and comparable data between BrS disease modeling studies, iPSC-CM populations of comparable purity, subtype and maturity should be used. On top of that, genetic alterations induced during reprogramming and/or passaging of iPSCs in culture create additional variability in the model, calling for the replication of obtained phenotypic and functional results in several (clonal) cell lines. Here, the scalability of iPSC-CMs in combination with high-throughput phenotyping techniques clearly present an advantage. In all of the cited BrS iPSC-CM reports, data from several differentiation rounds were combined, which is good practice. De la Roche and colleagues even specifically showed that the results between differentiation experiments were not significantly different and could be indeed pooled together [56]. Additionally, in the report of Selga et al., differences between iPSC-CM differentiation methods were addressed, and the effect of the mutation was shown to be similar in magnitude in iPSC-CMs derived using an EB- as well as a monolayer-based approach [69]. Selga et al. reported no significant differences between two clones differentiated from the same cell line. In most of the articles, several clones of patient and/or control iPSC-CMs were tested, but often not all were used for all comparisons. In fact, best practice to obtain rigid data from the models would require at least three different iPSC clones to be used for each patient and control individual and should be compared for all characteristics. Additionally, the use of multiple patient and control iPSC-CM lines is recommended. This should account for variability between individuals with different genetic backgrounds and comorbidities. In about half of the discussed reports, a single control iPSC-CM line was used [69,70,71,74,75,76], while in five articles two [55,56,66,67,68] and in two articles three [72,73] controls were analyzed. In some cases, authors observed differences in electrophysiological properties between their control cell lines [55,68] and then tended to compare their results to each control line separately. In other cases, where similar discrepancies between the control lines were observed (e.g., APD50 or dV/dT max values) [56,72,73], the authors compared their results from patient iPSC-CMs to the pooled range of values from all characterized control iPSC-CMs. It is clear that standardization, with the use of multiple clones and multiple individuals for all functional characterization studies, is needed in the field of iPSC-CM modelling.

Another important note is that the current iPSC-CM models only provide a cellular system demonstrating abnormal electrical behavior or structural abnormalities in single cells or 2D sheets, and caution must be exercised in extrapolating findings to the whole heart. The current iPSC differentiation protocols predominantly generate ventricular-like iPSC-CMs [40,41] but protocols aimed at specifically generating atrial-like [122] and nodal-like [123] iPSC-CMs have also been described. The co-culture of a controlled mixture of these different cell-types, potentially including non-cardiomyocytes (such as cardiac fibroblasts) in 3D tissues with proper extracellular matrix is predicted to be relevant to the study of cardiac arrhythmias. Though it will be hard to get these tissues organized in layers of endocardium and epicardium and/or attribute left- or right sidedness, study of arrhythmic events such as re-entry, conductance problems, local AP and current heterogeneity and even tissue fibrosis and structural changes that could all be relevant to the BrS phenotype and disease pathomechanisms should be feasible. In addition, such culture systems are very likely to improve on the maturity of the iPSC-CMs. Although quite some effort has already been made to create engineered heart tissue (EHT), 3D cardiac microtissues, organoids or microfluidic human-body-on-a-chip systems [48,124,125,126,127,128,129], this field needs further development and application of these novel, more physiological iPSC-CM based models. The latter will certainly provide interesting improvements for BrS disease modelling, as well as for the application of these models for drug testing, precision medicine and cardiotoxicity screening [130,131,132].

In conclusion, the thirteen discussed studies have proven the capability of the iPSC-CM models to recapitulate the BrS patient phenotype (Figure 2). However, no particularly compelling novel disease insights for BrS have emerged yet and it is important to take the current limitations of iPSC-CMs into account when using them as a model in studying Brugada syndrome and other cardiac arrhythmias. Focus on technologies to improve iPSC-CM maturity and create proper engineered heart tissues in combination with standardization of the experimental set-ups and more in-depth (functional) studies of the generated iPSC-CM models will help bridge the gap between model and clinical practice (Figure 2). Taken together, this exciting approach clearly holds great promise for BrS and cardiac disease research.

## Figures and Tables

**Figure 2 ijms-22-02825-f002:**
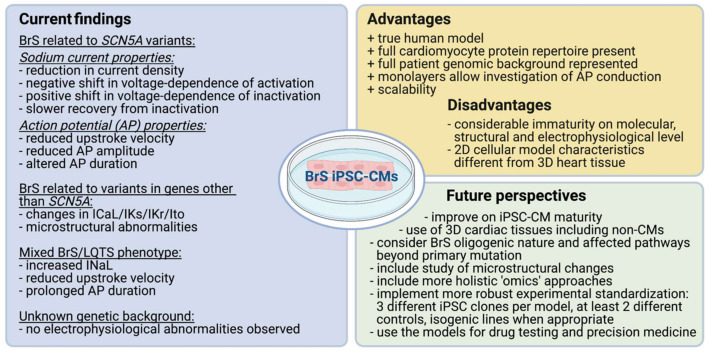
Summary of the main findings from the reviewed BrS iPSC-CM models, advantages and disadvantages of the model, as well as future perspectives for the use of iPSC-CMs in BrS research. (Generated with Biorender.com March 2021).

## Data Availability

Not applicable.

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
