# Peer review of "iPSC-Cardiomyocyte Models of Brugada Syndrome—Achievements, Challenges and Future Perspectives"

_ijms, 2021, doi:10.3390/ijms22062825_

Round 1
Reviewer 1 Report
Nijak et al. had an effort to summarize total 13 previous reports of iPSC-CMs models of the Brugada syndrome in this decade and discussed the resulting electrophysiological findings. Most of these iPSC-CMs models were mimicking the human Brugada phenotype caused by genetic mutations in Na channel genes. All of these had shown reduced INa, which had already been demonstrated in other expression system and pharmacological animal tissue models. Moreover, as the authors suggested, an oligogenic nature of the Brugada syndrome may take it much harder to reveal the detailed mechanisms of the syndrome. The iPSC-CMs have significant limitation for the Brugada phenotype in their maturity. The authors had well summarized in detail of the previous papers focusing on the Brugada syndrome using iPSC-CMs, thus the reviewer has only few concerns for this review paper, but if possible, the authors had better add one summary figure which can easily be understood the current and future perspective and limitation of the iPSC-CMs in the Brugada syndrome.
Author Response
Dear Reviewer,
Thank you for taking time to review our manuscript. We agree with the provided comments. As suggested, we included an overview figure (Figure 2, see below) summarizing the main findings from the reviewed Brugada syndrome iPSC-CM models, advantages and disadvantages of the models, as well as future perspectives that will be of interest in further iPSC-CM Brugada modelling studies. In the current manuscript version, we refer to this Figure 2 in the first and last paragraph of “5. Conclusions and future perspectives”.
Reviewer 2 Report
This is a comprehensive review of iPS-derived cell models of Brugada Syndrome.
It considers 13 recent papers using this cell model approach and provide a concise table summarizing the most important findings of those papers. This piece of information is greatly appreciated.
I consider that the contribution of this review should rely on the exhaustive analysis of those thirteen papers rather than repeated and, at times, inaccurate descriptions of methodologies extensively reviewed elsewhere. For instance, the scheme followed by the authors in the methodological section closely resembles the scheme used by Garg et al., from reference #61. It should probably be better if the authors briefly summarize the current methodological approaches, with advantages and limitations, and refer to previously published literature rather than replicating already reviewed information. As an example, in line 201 the authors refer to patch clamp as the “only” technique to measure actual membrane potential. This is misleading, the very ref. #61 clearly states that: “Sharp electrodes penetrating the cell membrane can accurately record MP and thus characterize APs in detail”. Also, the use of jargon like “in voltage-clamp” or “in current-clamp” should be avoided if no further explanation is provided for those methods.
In its present form, most of the descriptions of the 13 reviewed articles are too detailed, what ends up being a difficult-to-follow list of facts of one paper after the other. The authors should consider a more integrative and articulated approach where facts are interpreted rather than simply listed.
In the Conclusions section the authors should summarize what can be learned from all these papers and provide their own view. So far it appears that iPS-CM are a "powerful" model in studying BrS, whilst, at the same, time they do not provide novel compelling disease insights. This sounds highly contradictory and confuses the reader.
Minor comments
Line 275. There is a typo here, it should read “from” instead of “form”
Line 730. This consideration is a bit broad. It does not add any specific value.
Supplementary table 1 is difficult to read in its present format. Please minimize the use of different text orientations
Author Response
Dear Reviewer,
Thank you for taking time to review our manuscript. Please find our point-by-point answers below."I consider that the contribution of this review should rely on the exhaustive analysis of those thirteen papers rather than repeated and, at times, inaccurate descriptions of methodologies extensively reviewed elsewhere. For instance, the scheme followed by the authors in the methodological section closely resembles the scheme used by Garg et al., from reference #61. It should probably be better if the authors briefly summarize the current methodological approaches, with advantages and limitations, and refer to previously published literature rather than replicating already reviewed information. As an example, in line 201 the authors refer to patch clamp as the “only” technique to measure actual membrane potential. This is misleading, the very ref. #61 clearly states that: “Sharp electrodes penetrating the cell membrane can accurately record MP and thus characterize APs in detail”. Also, the use of jargon like “in voltage-clamp” or “in current-clamp” should be avoided if no further explanation is provided for those methods."
We thank the reviewer for pointing out the need to revise our methodological section. In the current manuscript version we adapted and shortened the electrophysiological methods, with more focus on the advantages and disadvantages of the selected techniques.
"In its present form, most of the descriptions of the 13 reviewed articles are too detailed, what ends up being a difficult-to-follow list of facts of one paper after the other. The authors should consider a more integrative and articulated approach where facts are interpreted rather than simply listed."
We appreciate the reviewer’s suggestion about our detailed descriptions of the reviewed articles, however we specifically chose to include a more detailed overview of each paper, as we aimed to mention important experimental details (e.g. number of used samples/clones, specific electrophysiological recording conditions) to draw the reader’s attention to these more subtle points, that can have a major impact on the obtained results. In this format, the aforementioned facts were also interpreted and addressed in our extensive discussion.
"In the Conclusions section the authors should summarize what can be learned from all these papers and provide their own view. So far it appears that iPS-CM are a "powerful" model in studying BrS, whilst, at the same, time they do not provide novel compelling disease insights. This sounds highly contradictory and confuses the reader."
We agree this can be confusing to the reader and have changed the sentence: “In fact, from the thirteen discussed studies, no particularly compelling novel disease insights emerged, but they have proven the capability of the iPSC-CM models to recapitulate the BrS patient phenotype” into two sentences (now starting with the achievement) that read as follows: “In fact, the thirteen discussed studies have proven the capability of the iPSC-CM models to recapitulate the BrS patient phenotype (Figure 2). No particularly compelling novel disease insights for BrS have emerged yet, but we are convinced further studies will contribute to this.”
Now in the whole paragraph we start with stating that iPSC-CMs are a powerful model and that the currently performed studies have shown that also for Brugada syndrome they can recapitulate the patient phenotype. We continue with the message that for now the models did not teach us anything completely novel about BrS pathomechanisms, but with improved technologies and more in-depth studies we are convinced that the approach holds great promise for BrS and cardiac disease research.
In the current manuscript version, we included a new overview figure (Figure 2) summarizing the main findings from the reviewed Brugada syndrome iPSC-CM models, advantages and disadvantages of the models, as well as future perspectives that will be of interest in further iPSC-CM Brugada modelling studies. We refer to this Figure 2 in the first and last paragraph of “5. Conclusions and future perspectives”.
"Minor comments"
Line 275. There is a typo here, it should read “from” instead of “form”
We have made the change. The new sentence reads: “Findings from published BrS iPSC-CM models.”
Line 730. This consideration is a bit broad. It does not add any specific value.
We thank the reviewer for pointing this out. The sentence: “These are potential confounders important to be taken into account when studying BrS iPSC-CM models.” has been removed.
Supplementary table 1 is difficult to read in its present format. Please minimize the use of different text orientations
We have adjusted the format of Supplementary table 1.